

# Effects of plyometric training on measures of physical fitness in racket sport athletes: a systematic review and meta-analysis

Nuannuan Deng[1], Kim Geok Soh[1], Borhannudin Abdullah[1] and Dandan Huang[2]

[1] Faculty of Educational Studies, Universiti Putra Malaysia, Selangor, Malaysia
[2] College of Physical Education, Chongqing University, Chongqing, China

## ABSTRACT

**Background:** Over the past decade, the popularity of racket sports has surged. Plyometric training (PT) has been the focus of extensive research because of the proven benefits it provides to athletes. However, there is a lack of systematic reviews and meta-analyses specifically evaluating the impact of PT on physical fitness metrics in racket sport athletes. This study aimed to conduct a comprehensive review and analysis of evidence derived from randomized controlled trials (RCTs) to evaluate the effects of PT on measures of physical fitness among racket sports athletes.

**Methods:** The electronic databases PubMed, Web of Science, SCOPUS, and SPORTDiscus were systematically searched up to June 2023 without placing any restrictions on the publication dates. The PICOS method was adopted to establish the inclusion criteria: (a) healthy athletes who participate in racket sports; (b) a PT program; (c) a control group; (d) assessment of physical fitness components pre- and post-PT; and (e) RCTs. The records' methodological quality was assessed utilizing the Physiotherapy Evidence Database (PEDro) scale. The certainty in the evidence related to each outcome was evaluated using the Grading of Recommendations Assessment, Development, and Evaluation (GRADE) assessment. A random-effects model was used to calculate effect sizes (ES; Hedges' g) between experimental and control groups.

**Results:** There were 14 eligible studies of moderate-to-high-quality, involving 746 athletes in total. The results revealed small-to-moderate effects ($p < 0.05$) of PT on muscle power (ES = 0.46), muscle strength (ES = 0.50), sprint speed (ES = 0.45), change of direction ability (ES = 0.76), and reaction time (ES = 0.67), while no clear evidence was found on balance and flexibility. The training-induced changes in muscle power showed no significant difference ($p > 0.05$) between youth (ES = 0.72) and adults (ES = 0.40). There were also similar muscle power improvements (ES = 0.36–0.54 *vs* 0.38–0.56, all $p > 0.05$) for a length of ≤7 weeks with ≤14 total PT sessions *vs* >7 weeks with >14 total PT sessions, and ≤2 weekly sessions *vs* >2 sessions. No adverse effects were reported in the included studies regarding the PT intervention. The certainty of evidence varied from very low to moderate.

Conclusions: Our findings demonstrated that PT has positive effects on important indices of physical fitness among athletes participating in racket sports. Future studies are required to clarify the optimal doses and examine interactions among training variables to further promote the physical fitness of this specific population.

Corresponding authors
Nuannuan Deng,
dengnuannuan117@gmail.com
Kim Geok Soh, kims@upm.edu.my

## INTRODUCTION

Racket sports have gained worldwide popularity and continue to grow in prominence (*Wörner & Safran, 2021*). The major racket sports include badminton, racquetball, padel, tennis, table tennis, and squash (*Cádiz Gallardo et al., 2023*). In these games, players utilize a handheld racket to propel a projectile back and forth, aiming to strategically position it in a way that prevents their opponent from successfully returning it (*Lees, 2003*). Moreover, these games involve a defined playing area in which the projectile must remain and a certain obstacle that it must surpass during each exchange (*Lees, 2003*). Success in racket sports heavily relies on both aerobic and anaerobic capacity, as players need a combination of quick reflexes, anticipation, change of direction ability, speed, and flexibility to swiftly reach the ball and avoid errors (*Fernandez, Mendez-Villanueva & Pluim, 2006*; *Edel et al., 2019*; *Farley et al., 2020*). A solid physical foundation provides greater opportunities for the development of psychological, tactical, and technical skills (*Bompa & Haff, 2009*; *Farley et al., 2020*). For example, to effectively return the shuttlecock across the entire badminton court, players must execute a wide range of movements, such as swift changes in direction, jumps, rapid arm actions, and lunges from different postural positions (*Phomsoupha & Laffaye, 2015*). Therefore, athletes need to possess not only significant muscular power and strength but also the ability to change direction quickly and maintain dynamic balance. Due to the advantages offered by enhanced physical fitness, a considerable amount of research has been dedicated to improving fitness metrics through various training methods (*Ozmen & Aydogmus, 2016*; *Falch, Guldteig Rædergård & van den Tillaar, 2019*; *Thiele et al., 2020*; *Nuñez et al., 2022*; *Cid-Calfucura et al., 2023*; *de Villarreal et al., 2023*; *Deng et al., 2023*). Among these methods, plyometric training (PT) is frequently recommended by researchers (*Oxfeldt et al., 2019*; *Eraslan et al., 2021*; *Ramirez-Campillo et al., 2022, 2023*).

The defining characteristic of PT is the stretch-shortening cycle (SSC), which includes an eccentric movement (lengthening action) followed by a concentric movement (shortening action) (*Heinecke, 2021*). The majority of the PT mechanism consists of two integral components. The first component involves converting the elastic energy accumulated during muscle stretching into power output during concentric contractions (*Wilk et al., 1993*). The second component senses muscular tension and length by using the proprioceptor signals generated during muscle stretching (*Davies, Riemann & Manske, 2015*). These sensory signals then transmit nerve impulses to the spinal cord, relaying information to alpha motor neurons responsible for activating agonist muscles, recruiting motor units, and inhibiting the contraction of antagonist muscles (*Markovic & Mikulic, 2010*). As mentioned above, PT induces a wide range of biomechanical and physiological adaptations. These adaptations make PT a valuable and advantageous training method for enhancing sports performance among athletes (*Cherni et al., 2021*; *Duchateau & Amiridis, 2023*). The significance of PT has been established through a growing body of scientific research showcasing its efficacy in enhancing various physical fitness factors, such as
muscle power, sprint speed, and change of direction ability, regardless of age or sex (*Ramirez-Campillo et al., 2020a*, *2020b*, *2021a*). Nevertheless, the aforementioned research encompassed athletes participating in various non-racket sports, such as soccer, volleyball, and basketball. It is crucial to acknowledge that the impact of PT may vary depending on the players' specific athletic background. As highlighted by *Lees (2003)*, racket sports possess distinct physical demands encompassing various fitness components. Hence, it may not be appropriate to directly apply the findings of these studies to athletes involved in racket sports (*Thapa et al., 2021*; *Sole et al., 2021*; *Ramirez-Campillo et al., 2021b*). Accordingly, the specific impact of PT on racket sports players' physical fitness still awaits discovery.

A systematic review with a meta-analysis can reveal the limitations and gaps within the existing literature on PT. This comprehensive analysis would provide valuable insights for scientists and practitioners. It would also inform them about potential directions for future studies (*Ramirez-Campillo et al., 2022*). However, there has yet to be a comprehensive review that conducts a meta-analysis of the publications regarding the impact of PT on the physical fitness of racket sports athletes. Therefore, this meta-analysis aims to fill the gap by analyzing the distinct effects of PT compared to control conditions on various components of physical fitness that are relevant to racket sport athletes. Concerning prior meta-analyses (*Sole et al., 2021*; *Deng et al., 2022*), we hypothesized that PT would yield greater effects than a control condition in enhancing the physical fitness of athletes involved in racket sports.

## METHODS

The research team of this study adhered to the guidelines outlined in the updated Preferred Reporting Items for Systematic Reviews and Meta-Analysis (PRISMA) statement (*Page et al., 2021*). This protocol was preregistered (registration number: CRD42023432050) at the International Prospective Register of Systematic Reviews (PROSPERO).

### Search strategy

Four databases were accessed for the search: PubMed, Web of Science, SCOPUS, and SPORTDiscus, and there were no limitations placed on publication dates up until June 2, 2023. Previous reviews (*Ramirez-Campillo et al., 2021a*; *Cádiz Gallardo et al., 2023*) were adopted to set up our search strategy, which was carried out by employing the Boolean operators "AND" and "OR". The Boolean search syntax displayed below was applied: "plyometric training" OR "ballistic training" OR "jump training" OR "plyometric exercise*" OR "power training" OR "stretch-shortening cycle" AND "racquet sport*" OR "racket sport*" OR "racket player*" OR "badminton" OR "squash" OR "padel" OR "tennis" OR "table tennis" OR "ping pong" OR "racquetball". Moreover, an extensive manual search was performed on both Google Scholar and reference lists from all identified articles to ensure that no pertinent articles were overlooked. Skilled librarians assisted in the data-gathering process to ensure accuracy and completeness. The detailed search strategy is available in Appendix S1. In addition, we reached out to authors to acquire full-text articles or any missing information, such as pre-test and post-test data.

## Inclusion and exclusion criteria

Following the PICOS method (*Liberati et al., 2009*), we selected the inclusion criteria by (a) population: healthy racket sport players (*e.g.*, tennis, table tennis); (b) intervention: applied a PT intervention lasting >2 weeks, including lower body exercises (*e.g.*, jumping, hopping, skipping) and/or upper body exercises (*e.g.*, medicine ball exercise, push-up) utilizing the SSC; (c) comparator: included a control group; (d) outcome: included at least one of the performance indicators related to physical fitness (*e.g.*, handgrip strength, straight-line sprint test <50 m); and (e) study design: randomized controlled trial.

Studies were excluded if they (a) involved injured athletes (*e.g.*, ankle sprain); (b) had interventions lasting ≤2 weeks; (c) did not provide adequate results (*e.g.*, mean and standard deviation); (d) investigated the effects of PT mixed with other training approaches (*e.g.*, balance training), to avoid contamination of the PT effects from other interventions; (e) tested the effects of PT without a control group; and (f) were conducted in languages other than English. In view of translation difficulties and the fact that most of the literature on PT is in English (*Ramirez-Campillo et al., 2018*), only English language studies were included.

## Study selection and data collection process

During the study's selection procedure, the retrieved publications were initially filtered for duplicates using specialist software (EndNote X9; Clarivate Analytics, London, UK). Following the removal of duplicates, we performed an in-depth review of relevant articles' titles, abstracts, and, finally, entire texts. Two researchers (ND and DH) worked independently throughout every step of the procedure, and any disagreements among the researchers were settled by consensus. Excluded full-text articles were noted, as well as the reasons for their exclusion.

## Data items

The data items were common metrics of physical fitness, including but not restricted to (a) muscle power (*e.g.*, jump height), (b) sprint speed (*i.e.*, 0–50 m), (c) muscle strength (*e.g.*, handgrip), (d) change of direction ability (*e.g.*, t-test (without a stimulus)), (e) flexibility (*e.g.*, sit and reach test), and (f) balance (*e.g.*, Y-balance test). Additionally, apart from the mentioned data elements, descriptive characteristics of both the PT interventions (*e.g.*, length, frequency) and the participants (*e.g.*, sex, sport) were extracted, and adverse effects were recorded. To conduct the meta-analysis, we specifically chose original articles that provided data suitable for calculation and utilized consistent outcome measures.

## Methodological quality

The methodological quality of each study included in the analysis was evaluated using the PEDro scale (*Maher et al., 2003*). There are 11 items on the PEDro checklist, but the first item was not assigned a rating. As a result, the scale used in the evaluation process had a maximum score of 10 and a minimum score of 0. Consistent with previous meta-analyses in the field of PT (*Stojanović et al., 2017*; *Ramirez-Campillo et al., 2021c*), the quality

evaluation was interpreted as follows: ≤3 points = poor quality, 4–5 points = moderate quality, 6–10 points = high quality. If trials have been reported on the PEDro database or relevant review articles, the supplied ratings were used in the assessment. Two independent reviewers (ND and DH) conducted the methodological quality assessment. In cases where there were discrepancies in the ratings between the reviewers, a third reviewer was involved to facilitate discussion and reach a consensus (KGS).

## Certainty of evidence

Two researchers (ND and DH) adopted the Grading of Recommendations, Assessment, Development, and Evaluation (GRADE) to evaluate the certainty of evidence (*Guyatt et al., 2011*; *Zhang et al., 2019a*, *2019b*). Initially, the evidence for each outcome exhibited a high level of certainty. However, it was subsequently downgraded on account of the factors outlined below: (a) risk of bias in studies: if the median PEDro scores were determined to be moderate (<6), the judgments were downgraded by one level; (b) indirectness: the specificity of populations, interventions, comparators, and outcomes ensured by the eligibility criteria automatically granted a default attribution of a low risk of indirectness; (c) risk of publication bias: if there were indications of publication bias, the judgments were downgraded by one level; (d) inconsistency: if the statistical heterogeneity ($I^2$) was determined to be high (>75%), the judgments were downgraded by one level; (e) imprecision: one level of downgrading was applied when a comparison involved either fewer than 400 participants or had a wide confidence interval (CI) around the effect estimate (*Zeng et al., 2022*); if both conditions were present, the certainty of the evidence was downgraded by two levels. In addition, if there were insufficient comparison trials to conduct a meta-analysis, the certainty of the evidence was rated as very low (*Ramirez-Campillo et al., 2022*).

## Statistical analyses

Meta-analyses were performed when a minimum of three studies provided sufficient data to calculate effect sizes (ES) (*Deng et al., 2022*; *Ramirez-Campillo et al., 2023*). Mean and standard deviation data from pre- and post-intervention measures were used to compute effect sizes for performance outcomes in both the PT and control groups (*i.e.*, Hedges' g). Data were standardized using post-intervention standard deviation values. A random-effects model for meta-analysis was utilized to analyze the pooled data. Additionally, a fixed-effects model was computed and presented to evaluate analytical robustness (*Higgins et al., 2003*). The values of ES were accompanied by 95% confidence intervals, and the calculated ES values were interpreted using the following scale: trivial (ES < 0.2), small (0.2 ≤ ES ≤ 0.6), moderate (0.6 < ES ≤ 1.2), large (1.2 < ES ≤ 2.0), very large (2.0 < ES ≤ 4.0), and extremely large (ES > 4.0) (*Hopkins et al., 2009*). In trials involving multiple intervention groups, the sample size of the control group was split proportionately so that all subjects could be compared (*Higgins, Deeks & Altman, 2008*). In cases where authors did not submit adequate data (in graphics or missing), we attempted to contact the corresponding authors. The investigation's outcome was left out of the analysis if the authors did not answer our inquiries or were unable to supply the

needed data. Yet, if data was shown in a figure but no numerical data was supplied in the article or tables, we employed the Graph Digitizer software (Digitizelt, Braunschweig, Germany) to extract relevant data from figures or graphs (*Drevon, Fursa & Malcolm, 2016*). The $I^2$ statistic was utilized to evaluate heterogeneity; varying degrees of heterogeneity were classified as low, moderate, and high. Specifically, heterogeneity was considered low when it was 25% or below, moderate from 25% to 75%, and high when it exceeded 75% (*Higgins & Thompson, 2002*). The extended Egger's test was utilized to evaluate the studies' publication bias risk (*Egger et al., 1997*). Furthermore, a sensitivity analysis was performed when the results of Egger's test were statistically significant. In addition, if it was not feasible to statistically pool the data, the findings were presented in a narrative format.

Subgroup analyses were performed to evaluate the potential impact of moderator factors. Potential sources of heterogeneity that might affect the training outcomes were accounted for in the analyses. These sources were determined *a priori*: program length (weeks), training frequency (weekly sessions), and total number of training sessions. A median split approach was used to divide the participants (*Moran et al., 2017*, *2018*). The participant's sex and age (youth (<18 years old) *vs* adult (≥18 years old)) were also considered as moderator factors. Stratification of the meta-analyses was conducted for each of these factors, and a threshold of $p < 0.05$ was utilized as the significance level to determine statistical significance. The Comprehensive Meta-Analysis software (Version 3.0; Biostat, Englewood, NJ, USA) was used for all of the analyses.

# RESULTS

## Study selection

As illustrated in Fig. 1, the databases yielded a total of 185 documents, and an additional 20 studies were found through Google Scholar and references. After manually removing duplicates, there were 113 records left. These records' titles and abstracts were screened, leading to the identification of 33 articles eligible for full-text analysis. Following a careful evaluation of all texts, 19 articles were removed, remaining 14 studies satisfied all of the eligibility requirements for the systematic review and meta-analysis.

## Methodological quality

After applying the PEDro checklist, nine studies obtained a score of 4 or 5 points, indicating a moderate level of methodological quality. Additionally, five studies attained a score of 6–10 points, indicating a high level of methodological quality (Table 1).

## Certainty of evidence

Table 2 displays the GRADE analysis results. The GRADE evaluations indicated that the certainty of the evidence for the outcomes was very low to moderate.

## Study characteristics

Table 3 provides a detailed overview of the participants' characteristics and PT programs employed in the included studies. Appendix S2 contains the data utilized in the

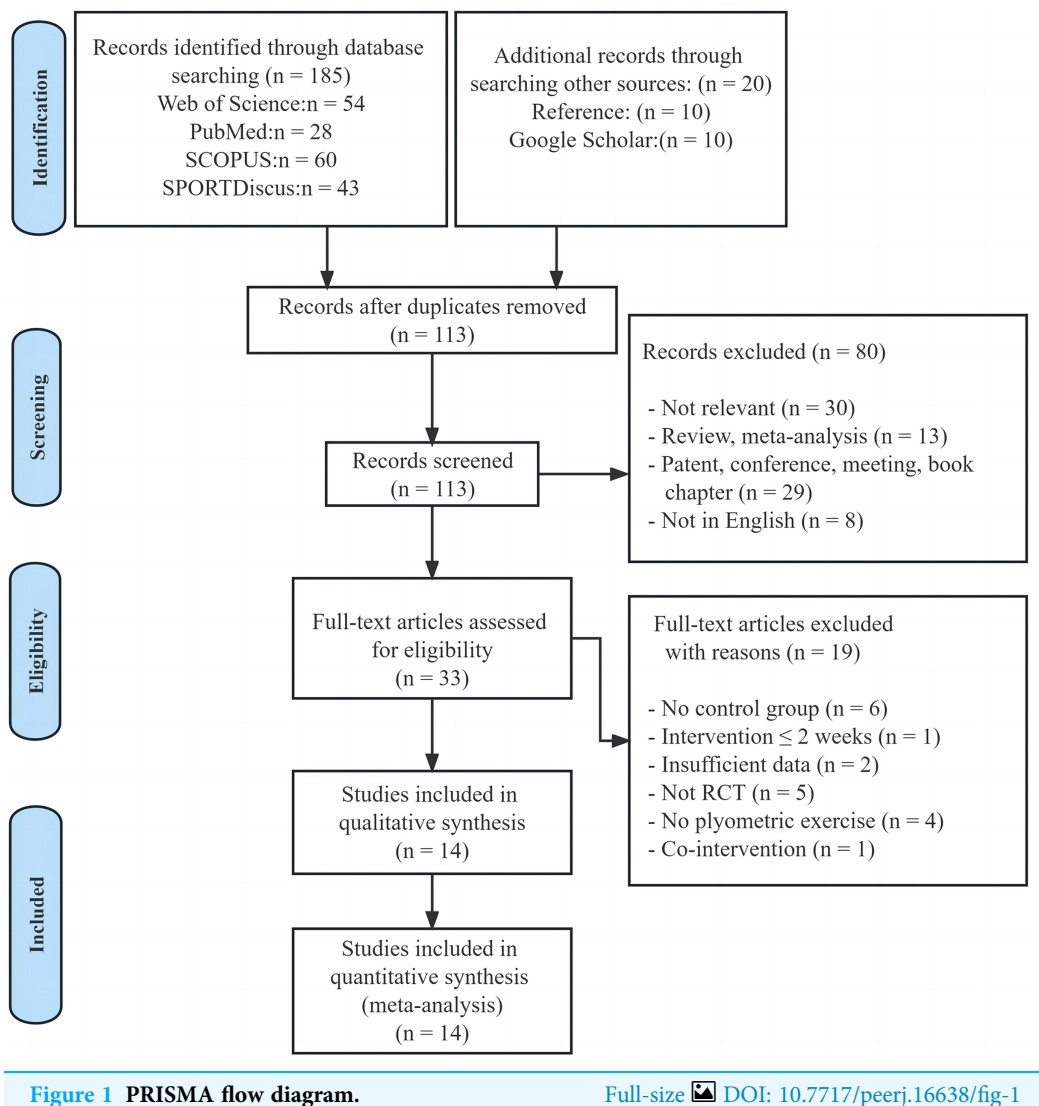

**Figure 1  PRISMA flow diagram.**               

meta-analyses. A total of 558 youth and adult athletes participated in the included studies. The intervention groups' sample sizes varied from 8 to 51 players, and their age range was between 12.5 and 25 years. The racket sports athletes included in the study were involved in badminton ($n = 8$), tennis ($n = 3$), and table tennis ($n = 3$).

In general, the intervention period of PT varied from 3 to 12 weeks, with an average duration of 6.6 weeks. The most common period was 8 weeks ($n = 7$). The number of training sessions each week varied from one to three. Overall, the most common training frequencies were two ($n = 6$) or three ($n = 7$) times per week. The total number of scheduled training sessions ranged from 3 to 36 sessions. The training volume ranged from 1 to 5 sets per exercise, with repetitions per set ranging from 7 to 30.

**Table 1  Physiotherapy Evidence Database (PEDro) scale ratings.**

| Study | N° 1 | N° 2 | N° 3 | N° 4 | N° 5 | N° 6 | N° 7 | N° 8 | N° 9 | N° 10 | N° 11 | Total* | Study quality |
|---|---|---|---|---|---|---|---|---|---|---|---|---|---|
| Alikhani et al. (2019) | 1 | 1 | 0 | 1 | 0 | 0 | 0 | 1 | 0 | 1 | 1 | 5 | Moderate |
| Albayati & Kaya (2022) | 0 | 1 | 0 | 1 | 0 | 0 | 0 | 1 | 0 | 1 | 1 | 5 | Moderate |
| Behringer et al. (2013) | 1 | 1 | 1 | 1 | 0 | 0 | 0 | 1 | 1 | 1 | 1 | 7 | High |
| Chou (2022) | 0 | 1 | 0 | 1 | 0 | 0 | 0 | 1 | 1 | 0 | 1 | 4 | Moderate |
| Chandra et al. (2023) | 1 | 1 | 0 | 1 | 0 | 0 | 0 | 1 | 1 | 1 | 1 | 5 | Moderate |
| Heang et al. (2012) | 0 | 1 | 0 | 1 | 0 | 0 | 0 | 1 | 0 | 1 | 1 | 5 | Moderate |
| Haghighi et al. (2021) | 0 | 1 | 0 | 1 | 0 | 0 | 0 | 1 | 1 | 1 | 1 | 6 | High |
| Kaabi, Mabrouk & Passelergue (2022) | 0 | 1 | 0 | 1 | 0 | 0 | 0 | 1 | 1 | 1 | 1 | 6 | High |
| Narang & Patil (2021) | 0 | 1 | 0 | 1 | 0 | 0 | 0 | 1 | 1 | 1 | 1 | 6 | High |
| Ozmen & Aydogmus (2017) | 1 | 1 | 0 | 1 | 0 | 0 | 0 | 1 | 0 | 1 | 1 | 5 | Moderate |
| Panda et al. (2022) | 1 | 1 | 0 | 1 | 0 | 1 | 1 | 1 | 1 | 1 | 1 | 8 | High |
| Rathore (2016) | 1 | 1 | 0 | 1 | 0 | 0 | 0 | 0 | 0 | 1 | 1 | 4 | Moderate |
| Salonikidis & Zafeiridis (2008) | 0 | 1 | 0 | 1 | 0 | 0 | 0 | 1 | 0 | 1 | 1 | 5 | Moderate |
| Zaferanieh et al. (2021) | 1 | 1 | 0 | 1 | 0 | 0 | 0 | 1 | 0 | 1 | 1 | 5 | Moderate |

Notes:
A detailed explanation for each PEDro scale item can be accessed at https://www.pedro.org.au/english/downloads/pedro-scale.
* From a possible maximal punctuation of 10.

**Table 2  GRADE analyses.**

| Outcomes | Certainty assessment | | | | | No of participants and studies | Certainty of evidence (GRADE) |
|---|---|---|---|---|---|---|---|
| | Risk of bias | Inconsistency | Indirectness | Imprecision | Risk of publication bias | | |
| Muscle power follow-up: range 3 to 12 weeks | Serious[a] | Not serious | Not serious | Serious[c] | Not serious | 350 (9 RCTs) | ⊕⊕◯◯ LOW |
| Muscle strength follow-up: range 8 to 12 weeks | Serious[a] | Not serious | Not serious | Serious[c] | Not serious | 155 (6 RCTs) | ⊕⊕◯◯ LOW |
| Sprint speed follow-up: range 3 to 12 weeks | Not Serious | Not serious | Not serious | Serious[c] | Serious[e] | 270 (6 RCTs) | ⊕⊕◯◯ LOW |
| Change of direction ability follow-up: range 3 to 12 weeks | Not serious | Serious[b] | Not serious | Serious[c] | Not serious | 324 (8 RCTs) | ⊕◯◯◯ VERY LOW |
| Reaction time follow-up: range 8 to 12 weeks | Not serious | Not serious | Not serious | Serious[c] | Not serious | 122 (4 RCTs) | ⊕⊕⊕◯ MODERATE |

Notes:
GRADE, grading of recommendations assessment, development and evaluation; RCTs, randomized controlled trials.
[a] Downgraded by one level due to average PEDro score being moderate (<6).
[b] Downgraded by one level due to the high impact of statistical heterogeneity (>75%).
[c] Downgraded by one level, as <400 participants were available for a comparison or a wide confidence interval (CI) around the effect estimate; we considered a CI to be wide if it included both a small (0.2–0.6) and large effect size (>1.2–2.0). Downgraded by two levels in case of imprecision based on both assessed points.
[e] Downgraded by one level (Egger's test $p < 0.05$).

**Table 3 Participant's characteristics and PT interventions in the included studies.**

| Study | Sport | Sex | n | Age | Level/experience | Intervention | PTG | CG | Test(s) | Outcome(s) |
|---|---|---|---|---|---|---|---|---|---|---|
| *Heang et al. (2012)* | Badminton | Mixed | 42 | 18–20 yrs | Collegiate EG: 2.0 ± 1.4 yrs CG: 1.4 ± 0.8 yrs | Freq: 1 time/week Time: NR Length: 3 weeks | Hops, jumps, bounds, hexagon drill 2–5 sets × 6–15 reps | College co-curriculum programme | Agility (Illinois test) | Illinois test ↑ |
| *Ozmen & Aydogmus (2017)* | Badminton | Mixed | 20 | 12.5 ± 0.2 yrs | NR At least 2 yrs | Freq: 2 times/week Time: NR Length: 6 weeks | Hops, jumps, bounds, hexagon drill 2–5 sets × 5–15 reps | Routine training | Power (SJ), agility (Illinois test) | SJ ↑, Illinois test ↑ |
| *Alikhani et al. (2019)* | Badminton | F | 22 | EG: 22.00 ± 1.30 yrs CG: 22.00 ± 0.84 yrs | Club EG: 2.5 ± 1.0 yrs CG: 3.0 ± 0.9 yrs | Freq: 3 times/week Time: 20 min Length: 6 weeks | Hops, jumps, bounds, triple broad 1–2 sets × 5–30 reps | Routine training | Balance (Y-balance test) | Y-balance test ↑ |
| *Narang & Patil (2021)* | Badminton | Mixed | 40 | 18–25 yrs | Club NR | Freq: 2 times/week Time: NR Length: 8 weeks | Ballistic six exercises 3 sets × 10–20 reps | Theraband exercises | Power (MBT), reaction time | MBT ↑, reaction time ↑ |
| *Panda et al. (2022)* | Badminton | Mixed | 30 | EG: 19.06 ± 1.33 yrs CG: 17.5 ± 0.52 yrs | State ≥2 yrs | Freq: 2 times/week Time: 35 min Length: 4 weeks | Jumps, hops, sit ups, step ups, box shuffle 2–5 sets × 5–15 reps | Routine training | Power (VJ), speed (30 m) agility (t-test), | VJ ↑, t-test ↑, 30 m ↑ |
| *Chou (2022)* | Badminton | NR | 16 | 19.8 ± 3.34 yrs | Collegiate NR | Freq: 3 times/week Time: 60–90 min Length: 8 weeks | Hops, jumps, side strides, rope ladder side touch 3 sets × 8–12 reps | Routine training | Power (VJ), speed (30 m), agility (6 × 4 m shuttle run) | VJ ↑, 30 m ↑, 6 × 4 m shuttle run ↑ |
| *Albayati & Kaya (2022)* | Badminton | NR | 21 | EG:13.43 ± 0.53 yrs CG:13.14 ± 0.69 yrs | NR | Freq: 3 times/week Time: 25–30 min Length: 12 weeks | Hops, jumps, jumping using arms | Elastic resistance training | Power (VJ), agility (t-test), Speed (10 m), reaction time, flexibility (SAR), strength (handgrip) | VJ ↑, t-test ↑, handgrip ↑, 10 m ↔, reaction time ↔, SAR ↔ |
| *Chandra et al. (2023)* | Badminton | M | 102 | 18–25 yrs | Collegiate ≥3 yrs | Freq: 2 times/week Time: 20 min Length: 3 weeks | Hops, jumps 2–5 sets × 6–15 reps | Routine training | Power (SBJ), speed (30 m), agility (t-test) | SBJ ↑, t-test ↑, 30 m ↑ |
| *Behringer et al. (2013)* | Tennis | M | 36 | 15.03 ± 1.64 yrs | Club Averaged 6.15 yrs | Freq: 2 times/week Time: 30–60 min Length: 8 weeks | Jumps, skips, hops, push-ups, medicine ball exercises 2–4 sets × 10–20 reps | Routine training | Strength (leg press, chest press) | Leg press ↑, chest press ↑ |

(Continued)

| Study | Sport | Participants | | | Level/experience | Intervention | Type of exercise | | Test(s) | Outcome(s) |
|---|---|---|---|---|---|---|---|---|---|---|
| | | Sex | n | Age | | | PTG | CG | | |
| Rathore (2016) | Tennis | M | 60 | 18–23 yrs | National, state, or inter-varsity NR | Freq: 3 times/week Time: 45 min Length: 8 weeks | Not described | Routine training | Agility (Illinois test) | Illinois test ↑ |
| Salonikidis & Zafeiridis (2008) | Tennis | M | 64 | 21.1 ± 1.3 yrs | Novice 2–3 yrs | Freq: 3 times/week Time: NR Length: 9 weeks | Jumps, hops 2 sets × 6–8 reps | Routine training | Power (DJ), strength (Fmax, speed (12 m), reaction time | DJ ↑, Fmax ↑, 12 m ↑, reaction time ↑ |
| Haghighi et al. (2021) | Table tennis | M | 30 | 24 ± 7 yrs | Provincial and national 7 ± 3 yrs | Freq: 3 times/week Time: NR Length: 8 weeks | Jumps, swimming launch, medicine ball exercises (3 kg) 3–4 sets × 8–30 reps | Routine training | Strength (Chest press, Leg extension) | Chest press ↑, Leg extension ↑ |
| Zafaranieh et al. (2021) | Table tennis | M | 30 | 24 ± 7 yrs | Elite Averaged of 5 yrs | Freq: 3 times/week Time: NR Length: 8 weeks | Jumps, swimming launch, medicine ball exercises (3 kg) 3–4sets × 8–30 reps | Routine training | Power (SJT, MBP), strength (handgrip), agility (t-test), reaction time | reaction time ↔, handgrip ↔, SLJ ↔, MBP ↔, t-test ↑ |
| Kaabi, Mabrouk & Passelergue (2022) | Table tennis | M | 45 | 16–17.4 yrs | Elite >3 yrs | Freq: 2 times/week Time: NR Length: 8 weeks | Jumps, medicine ball exercises 3–4 sets × 5–10 reps | Routine training | power (CMJ), strength (handgrip), speed (5 m), agility (t-test) | Handgrip ↑, CMJ ↑, 5 m ↑, t-test ↑ |

**Note:**
NR, not reported; yrs, years; Exp, sports experience; M, male; F, female; Freq, frequency; CG, control group; PTG, plyometric training group; reps, repetitions; CMJ, vertical countermovement jump; SBJ, standing broad jump; DJ, drop jump; VJ, vertical jump; MBT, medicine ball throw; SJT, Sargent jump test; SJ, squat jump test; MBP, medicine ball put; Fmax, maximum isometric force (knee extension); SAR, sit and reach test; ↑, significant within-group improvement; ↔, non-significant within-group.

## Study outcomes

### *Muscle power*

In terms of muscle power, data from six studies were analyzed, which included a total of nine experimental groups and six control groups (pooled $n = 350$). The results indicated a small effect of PT on muscle power (ES = 0.46; 95% CI [0.25–0.67]; $p < 0.001$; $I^2 = 0.0\%$; Egger's test $p = 0.413$; GRADE: low; Fig. 2). When a fixed-effect model was utilized, the pooled estimate remained consistent.

No significant sub-group differences ($p = 0.444$) were observed between ≤7 weeks with ≤14 total PT sessions (ES = 0.54; 95% CI [0.25–0.83]; within-group $I^2 = 20.6\%$, three study groups) and >7 weeks with >14 total PT sessions (ES = 0.38; 95% CI [0.08–0.68]; within-group $I^2 = 0.0\%$, seven study groups) of training. When a fixed-effect model was utilized, the pooled estimate remained consistent.

Similarly, no significant sub-group differences ($p = 0.364$) were noted between ≤2 sessions per week (ES = 0.36; 95% CI [0.06–0.66]; within-group $I^2 = 0.0\%$, five study groups) and >2 sessions per week (ES = 0.56; 95% CI [0.26–0.85]; within-group $I^2 = 0.0\%$, five study groups). When a fixed-effect model was utilized, the pooled estimate remained consistent.

Additionally, between youth (ES = 0.72; 95% CI [0.25–1.22]; within-group $I^2 = 0.0\%$, three study groups) and adult (ES = 0.40; 95% CI [0.17–0.63]; within-group $I^2 = 0.0\%$, seven study groups) racket sport participants, no significant sub-group differences ($p = 0.244$) were found. When a fixed-effect model was utilized, the pooled estimate remained consistent.

### *Muscle strength*

Data from six studies were gathered to evaluate muscle strength, encompassing seven experimental groups and six control groups (pooled $n = 155$). The results displayed a moderate effect of PT on muscle strength (ES = 0.50; 95% CI [0.19–0.82]; $p = 0.002$; $I^2 = 0.0\%$; Egger's test $p = 0.800$; GRADE: low; Fig. 3). When a fixed-effect model was utilized, the pooled estimate remained consistent.

### *Sprint speed*

Sprint speed was assessed based on data from six studies, comprising seven experimental groups and six control groups (pooled $n = 270$). Egger's test yielded a $p = 0.009$. After sensitivity analysis, the removal of one study (*Chandra et al., 2023*) permitted an Egger's test of $p \geq 0.05$. This led to the final consideration of five studies, which included six experimental and five control groups. PT had a small impact on sprint speed (ES = 0.45; 95% CI [0.15–0.75]; $p = 0.004$; $I^2 = 0.0\%$; GRADE: low; Fig. 4). When a fixed-effect model was utilized, the pooled estimate remained consistent.

### *Change of direction ability*

Data from eight studies were collected for the analysis of change of direction ability, consisting of nine experimental groups and six control groups (pooled $n = 324$). The results indicated a moderate effect of PT on change of direction ability (ES = 0.76; 95% CI [0.27–1.25]; $p = 0.003$; $I^2 = 76.5\%$; Egger's test $p = 0.389$; GRADE: very low; Fig. 5).

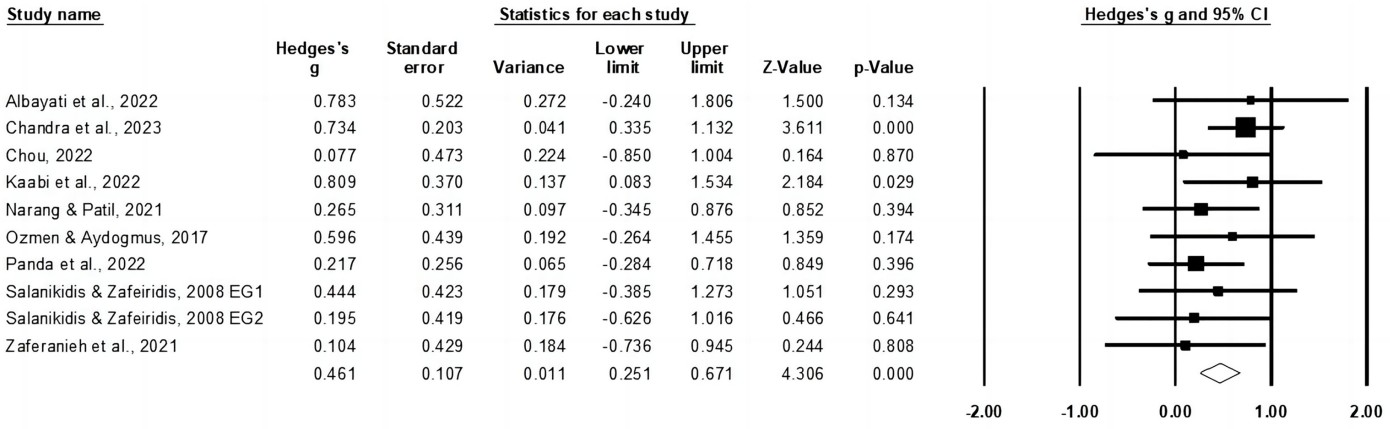

| Study name | Statistics for each study | | | | | | | Hedges's g and 95% CI |
|---|---|---|---|---|---|---|---|---|
| | Hedges's g | Standard error | Variance | Lower limit | Upper limit | Z-Value | p-Value | |
| Albayati et al., 2022 | 0.783 | 0.522 | 0.272 | -0.240 | 1.806 | 1.500 | 0.134 | |
| Chandra et al., 2023 | 0.734 | 0.203 | 0.041 | 0.335 | 1.132 | 3.611 | 0.000 | |
| Chou, 2022 | 0.077 | 0.473 | 0.224 | -0.850 | 1.004 | 0.164 | 0.870 | |
| Kaabi et al., 2022 | 0.809 | 0.370 | 0.137 | 0.083 | 1.534 | 2.184 | 0.029 | |
| Narang & Patil, 2021 | 0.265 | 0.311 | 0.097 | -0.345 | 0.876 | 0.852 | 0.394 | |
| Ozmen & Aydogmus, 2017 | 0.596 | 0.439 | 0.192 | -0.264 | 1.455 | 1.359 | 0.174 | |
| Panda et al., 2022 | 0.217 | 0.256 | 0.065 | -0.284 | 0.718 | 0.849 | 0.396 | |
| Salanikidis & Zafeiridis, 2008 EG1 | 0.444 | 0.423 | 0.179 | -0.385 | 1.273 | 1.051 | 0.293 | |
| Salanikidis & Zafeiridis, 2008 EG2 | 0.195 | 0.419 | 0.176 | -0.626 | 1.016 | 0.466 | 0.641 | |
| Zaferanieh et al., 2021 | 0.104 | 0.429 | 0.184 | -0.736 | 0.945 | 0.244 | 0.808 | |
| | 0.461 | 0.107 | 0.011 | 0.251 | 0.671 | 4.306 | 0.000 | |

Favours Control   Favours Plyometric

**Figure 2 Forest plot of changes in muscle power in racket sport athletes participating in training intervention compared to controls.** Values shown are effect sizes (Hedges' g) with 95% confidence intervals (CI). The size of the plotted squares reflects the statistical weight of the study. EG, experimental group *Albayati & Kaya (2022), Chandra et al. (2023), Chou (2022), Kaabi, Mabrouk & Passelergue (2022), Narang & Patil (2021), Ozmen & Aydogmus (2017), Panda et al. (2022), Salonikidis & Zafeiridis (2008), Zaferanieh et al. (2021).*

| Study name | Statistics for each study | | | | | | | Hedges's g and 95% CI |
|---|---|---|---|---|---|---|---|---|
| | Hedges's g | Standard error | Variance | Lower limit | Upper limit | Z-Value | p-Value | |
| Albayati et al., 2022 | 0.765 | 0.521 | 0.271 | -0.256 | 1.786 | 1.468 | 0.142 | |
| Behringer et al. 2013 | 0.516 | 0.401 | 0.161 | -0.270 | 1.302 | 1.286 | 0.198 | |
| Haghighi et al., 2021 | 0.478 | 0.446 | 0.199 | -0.396 | 1.351 | 1.072 | 0.284 | |
| Kaabi et al., 2022 | 1.251 | 0.390 | 0.152 | 0.486 | 2.016 | 3.205 | 0.001 | |
| Salanikidis & Zafeiridis, 2008 EG1 | 0.164 | 0.419 | 0.175 | -0.656 | 0.985 | 0.392 | 0.695 | |
| Salanikidis & Zafeiridis, 2008 EG2 | 0.116 | 0.418 | 0.175 | -0.704 | 0.936 | 0.278 | 0.781 | |
| Zaferanieh et al., 2021 | 0.188 | 0.429 | 0.184 | -0.654 | 1.029 | 0.437 | 0.662 | |
| | 0.502 | 0.162 | 0.026 | 0.186 | 0.819 | 3.111 | 0.002 | |

Favours Control   Favours Plyometric

**Figure 3 Forest plot of changes in muscle strength in racket sport athletes participating in training intervention compared to controls.** Values shown are effect sizes (Hedges' g) with 95% confidence intervals (CI). The size of the plotted squares reflects the statistical weight of the study. EG, experimental group *Albayati & Kaya (2022), Behringer et al. (2013), Haghighi et al. (2021), Kaabi, Mabrouk & Passelergue (2022), Salonikidis & Zafeiridis (2008), Zaferanieh et al. (2021).*

When utilizing a fixed-effect model, a somewhat higher pooled estimate was noted (ES = 0.85, 95% CI [0.62–1.08]; $p < 0.001$).

### Reaction time

Four trials with nine experimental groups and six control groups were used to derive data on reaction time (pooled $n$ = 122). The results showed a moderate effect of PT on reaction time (ES = 0.67; 95% CI [0.16–1.18]; $p$ = 0.010; $I^2$ = 46.6%; Egger's test $p$ = 0.172; GRADE:

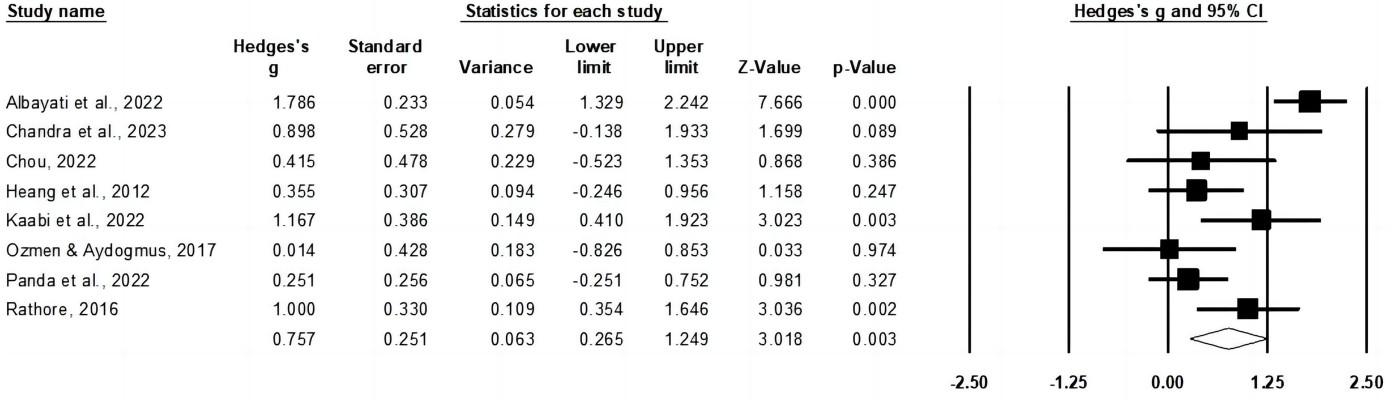

**Figure 4 Forest plot of changes in sprint speed in racket sport athletes participating in training intervention compared to controls.** Values shown are effect sizes (Hedges' g) with 95% confidence intervals (CI). The size of the plotted squares reflects the statistical weight of the study. EG, experimental group *Albayati & Kaya (2022)*, *Chou (2022)*, *Kaabi, Mabrouk & Passelergue (2022)*, *Panda et al. (2022)*, *Salonikidis & Zafeiridis (2008)*.

**Figure 5 Forest plot of changes in change of direction ability in racket sport athletes participating in training intervention compared to controls.** Values shown are effect sizes (Hedges' g) with 95% confidence intervals (CI). The size of the plotted squares reflects the statistical weight of the study *Albayati & Kaya (2022)*, *Chandra et al. (2023)*, *Chou (2022)*, *Heang et al. (2012)*, *Kaabi, Mabrouk & Passelergue (2022)*, *Ozmen & Aydogmus (2017)*, *Panda et al. (2022)*, *Rathore (2016)*.

moderate; Fig. 6). When utilizing a fixed-effect model, a slightly higher pooled estimate was observed (ES = 0.72, 95% CI [0.35–1.08]; $p < 0.001$).

### Balance and flexibility

Due to the limited number of studies, balance and flexibility were not incorporated in the meta-analysis. Thus, the certainty of the evidence was automatically rated as very low. A study revealed that the dynamic balance of badminton players significantly improved after 8 weeks of PT (*Alikhani et al., 2019*). However, another study measured flexibility among badminton players. This research demonstrated a non-significant change in the flexibility

| Study name | Statistics for each study | | | | | | | Hedges's g and 95% CI |
| --- | --- | --- | --- | --- | --- | --- | --- | --- |
| | Hedges's g | Standard error | Variance | Lower limit | Upper limit | Z-Value | p-Value | |
| Albayati et al., 2022 | 0.208 | 0.502 | 0.252 | -0.776 | 1.192 | 0.414 | 0.679 | |
| Narang & Patil, 2021 | 1.289 | 0.342 | 0.117 | 0.619 | 1.959 | 3.770 | 0.000 | |
| Salanikidis & Zafeiridis, 2008 EG1 | 0.820 | 0.435 | 0.189 | -0.032 | 1.672 | 1.887 | 0.059 | |
| Salanikidis & Zafeiridis, 2008 EG2 | 0.893 | 0.438 | 0.191 | 0.036 | 1.751 | 2.041 | 0.041 | |
| Zaferanieh et al., 2021 | -0.078 | 0.428 | 0.184 | -0.918 | 0.762 | -0.182 | 0.855 | |
| | 0.671 | 0.259 | 0.067 | 0.162 | 1.179 | 2.585 | 0.010 | |

**Figure 6 Forest plot of changes in reaction time in racket sport athletes participating in training intervention compared to controls.** Values shown are effect sizes (Hedges' g) with 95% confidence intervals (CI). The size of the plotted squares reflects the statistical weight of the study. EG, experimental group *Albayati & Kaya (2022)*, *Narang & Patil (2021)*, *Salonikidis & Zafeiridis (2008)*, *Zaferanieh et al. (2021)*.

assessment for the pre-and post-test of the experimental group after 12 weeks of PT (*Albayati & Kaya, 2022*).

### Adverse effects

The studies considered in this analysis did not document any cases of injury, soreness, pain, damage, fatigue, or adverse health effects linked to the implemented PT interventions.

## DISCUSSION

The present study is the first meta-analysis to look at the effects of PT on racket sports athletes. The findings indicated that PT yielded significant small-to-moderate gains on measures of physical fitness (*e.g.*, muscle power, reaction time) in racket sport players. In most cases, the results exhibited heterogeneity ranging from low to high levels ($I^2 = 0.0–76.5\%$). No substantial changes in mean effects or CIs were observed when employing both random- and fixed-effects models, demonstrating the robustness of the results across all meta-analyses. According to the GRADE assessment, the evaluated outcomes were found to have a very low to moderate level of evidence. Additionally, no definitive evidence was observed for balance and flexibility performance. Moderator analyses were performed for only muscle power due to the availability of at least three studies per moderator. Participants' age and training variables (*i.e.*, length, frequency, and total PT sessions) had no significant moderating effects on muscle power changes following PT.

### Muscle power

Power is a vital attribute of muscle function and holds significant importance for athletes participating in competitive sports (*Kraemer & Looney, 2012*; *Möck et al., 2018*). During matches and practice sessions, players consistently have to perform explosive actions, such as quick accelerations or decelerations, rapid changes of direction, and precise strokes

within limited time frames. These requirements emphasize the crucial role of power as a determining factor in achieving success (*Reid & Schneiker, 2008*; *Fernandez-Fernandez, Sanz-Rivas & Mendez-Villanueva, 2009*). Our results indicate significant increases in muscle power after the PT program. These findings align with previous meta-analyses that examined athletes from diverse team sports (*Slimani et al., 2016*; *Ramirez-Campillo et al., 2020b*, *2021a*). In sports such as table tennis, muscle power plays a vital role as it enables players to generate strong muscle contractions, facilitating the execution of various tasks such as exchanging shots with opponents (*Zagatto, Papoti & Gobatto, 2008*). Indeed, PT has the potential to enhance power production by improving coordination and neuromuscular adaptations (*e.g.*, synchronization, selective muscle activation, and increased recruitment of motor units) (*Mirzaei, Norasteh & Asadi, 2013*; *Chelly, Hermassi & Shephard, 2015*). In addition, the hypothesis that PT has a more significant impact on jump performance is frequently proposed due to its foundation in utilizing the SSC to enhance explosive actions. This claim has been supported by several studies (*Markovic, 2007*; *Ramírez-Campillo et al., 2015*; *Kubo, Ishigaki & Ikebukuro, 2017*).

Concerning subgroup analyses, there were no detected between-group differences in muscle power gains after ≤7 weeks (with ≤14 total sessions) *vs* >7 weeks (with >14 total sessions) or <2 sessions per week *vs* ≥2 sessions per week. Additionally, no significant differences were observed when comparing the effects between different age groups (youths *vs* adults). Similarly, previous meta-analyses on PT also did not detect significant differences between subgroups or correlations between training variables and changes in muscle power (*Sole et al., 2021*; *Deng et al., 2023*). Nevertheless, according to the chronological adaptation tempo, longer training periods with a higher number of sessions can provide more time for physiological mechanisms to enhance muscle power (*Markovic & Mikulic, 2010*; *Ramirez-Campillo et al., 2018*). Moreover, longer PT interventions enable a greater volume of exercises to be completed, resulting in more substantial enhancements in muscle power (*Ramírez-Campillo et al., 2015*; *Asadi et al., 2017*). From a practical standpoint, implementing a lower frequency of PT sessions may give players more time to focus on other crucial parts of their preparation (*Bouguezzi et al., 2018*). However, the current meta-analysis failed to identify any significant subgroup differences regarding PT variables. In addition, consistent with our findings, prior research has demonstrated that PT leads to enhancements in muscle power for athletes, regardless of their age (*Sole et al., 2021*; *Ramirez-Campillo et al., 2020b*, *2021a*). Collectively, the current evidence suggests that implementing a PT program can lead to notable enhancement in muscle power among racket sport athletes, irrespective of their age and training parameters.

## Muscle strength

Muscle strength is a vital factor that influences power and speed in athletic performance (*Chanavirut et al., 2017*). Our analysis revealed a notable enhancement in muscle strength following PT. These findings align with previous studies conducted on diverse population groups, which also reported that PT was an effective training modality for enhancing muscle strength (*de Villarreal, Requena & Newton, 2010*; *Ramirez-Campillo et al., 2021a*;

*Morris et al., 2022*). The strength improvements induced by PT can be attributed to neural adaptations, including enhancements in firing frequency, synchronization, and motor drive of motor units (*Markovic & Mikulic, 2010*). Furthermore, muscle hypertrophy may also explain PT-induced strength gains (*Grgic, Schoenfeld & Mikulic, 2020*). Of note, grip strength has frequently been utilized as a measure of muscular strength in both the current review (*Zaferanieh et al., 2021*; *Albayati & Kaya, 2022*; *Kaabi, Mabrouk & Passelergue, 2022*) and earlier research (*Trosclair et al., 2011*; *Watanabe et al., 2019*). In fact, grip strength is essential for executing various movements specific to racket sports, as players consistently engage in complex skills while firmly holding the racket (*Fett, Ulbricht & Ferrauti, 2020*; *Haryanto et al., 2021*). The effects of PT can vary depending on the nature of the training protocol (*Booth & Orr, 2016*). Greater improvements in grip strength are observed using PT with upper-body exercises (*e.g.*, medicine ball exercise) (*Zaferanieh et al., 2021*; *Kaabi, Mabrouk & Passelergue, 2022*). It is interesting to note that *Albayati & Kaya (2022)* conducted a study primarily utilizing lower-body PT (*e.g.*, jumps, hops) and observed positive results in terms of handgrip strength among young badminton players. Recent research has reported a significant association between handgrip strength and jump performance (*França et al., 2023*; *Maurya et al., 2023*), which may help explain the findings. Moreover, when analyzing and interpreting handgrip strength during childhood and adolescence, it is essential to consider the influence of biological maturation and growth, since changes in muscle mass and muscle fiber size may be the contributory factors (*Gómez-Campos et al., 2018*; *Yapici et al., 2022*; *Wind et al., 2010*).

In addition, one of the included articles observed that weightlifting is more effective than PT in improving muscle strength among table tennis players (*Kaabi, Mabrouk & Passelergue, 2022*). However, a recent meta-analysis comparing weightlifting training with PT indicated that both methods can potentially lead to similar improvements in muscle strength (*Morris et al., 2022*). The question of whether weightlifting training is indeed superior to PT still needs to be confirmed in future studies. From a practical perspective, muscle strength forms the foundation for numerous attributes linked to enhancing an individual's performance in a wide range of both general and sport-specific skills (*Suchomel, Nimphius & Stone, 2016*). This implication underscores the meaningful significance of training interventions designed to enhance muscle strength, where PT offers several advantages for attaining this goal.

## Sprint speed

In racket sports, the ability to quickly respond to actions executed by the opponent is crucial, and sprint speed plays a significant role in this regard (*Fernandez-Fernandez et al., 2016*). Following PT, there was a noticeable increase in sprint speed in the present analysis. The findings align with previous meta-analyses conducted on athletes from other sports, such as volleyball and soccer (*van de Hoef et al., 2020*; *Ramirez-Campillo et al., 2020a*, *2021b*). Indeed, PT has practical implications for skilled athletes participating in sports involving initial accelerations, specific explosive actions, and short-distance and/or high-intensity sprints (*de Villarreal, Requena & Cronin, 2012*). Improved sprint performance following PT can be attributed to several factors, including neural factors

(*e.g.*, increased muscle activation level and stimulation of spinal reflex pathways) (*Komi & Gollhofer, 1997*; *Taube, Leukel & Gollhofer, 2012*) and physiological factors (*e.g.*, increased muscle force per fiber and cross-sectional area) and/or muscle-tendon unit adaptations (*e.g.*, enhanced effectiveness in storing and releasing elastic energy) (*Markovic & Mikulic, 2010*). Taken together, these factors may improve SSC efficacy (*Ramirez-Campillo et al., 2021c*). The enhanced SSC efficacy in the lower body musculature is expected to lead to increased force generation during the concentric phase following a rapid eccentric muscle action, which is a crucial factor for increasing sprint speed (*Rimmer & Sleivert, 2000*; *Ronnestad et al., 2008*; *de Villarreal, Requena & Cronin, 2012*; *Hammami et al., 2016*). Moreover, maximum sprint speed is significantly correlated with lower-extremity strength and power (*Harris et al., 2008*; *Peñailillo et al., 2016*). Improvements in both lower-extremity strength and power have been reported after PT (*Ramírez-delaCruz et al., 2022*; *Deng et al., 2023*).

Furthermore, some researchers highlight that training programs incorporating greater horizontal acceleration (*i.e.*, horizontal jumping and skipping) can optimize gains in sprint performance (*de Villarreal, Requena & Cronin, 2012*). Particularly, in shorter distances (*e.g.*, ≤10 m), the application of horizontal force on the ground is crucial, making a higher PT load in the horizontal direction likely to yield greater benefits during the initial acceleration phase (*Chaabene et al., 2021*; *Ramirez-Campillo et al., 2020a*, *2021a*). When approaching high speeds, PT with a greater focus on vertical movement may result in greater benefits, especially after vertical workouts with a higher rate of force buildup and shorter ground contact times (*Ramirez-Campillo et al., 2020a*, *2021c*). It should be noted that the bulk of trials included in this meta-analysis used mixed PT regimens, which encompassed both vertical and horizontal drills. This combination of training modalities may contribute to the observed enhancements in linear sprint speed. These observations reinforce the value of PT programs to improve the sprint speed of athletes in racket sports.

## Change of direction ability

Change of direction ability is a crucial motor skill essential for success in sports (*Brughelli et al., 2008*). Compared to control conditions, our findings indicate that PT improves change of direction ability in athletes involved in racket sports. Improvement in change of direction ability through PT is supported by several previous meta-analyses (*Asadi et al., 2016*; *Falch, Guldteig Rædergård & van den Tillaar, 2019*; *Pardos-Mainer et al., 2021*; *Ramirez-Campillo et al., 2021a*). Several neuromuscular adaptations may explain the improved change of direction ability after PT, such as increased firing frequencies and motor unit recruitment (*Aagaard et al., 2002*; *Arazi, Coetzee & Asadi, 2012*).
The physiological changes resulting from these adaptations may lead to a faster rate of force generation and power output. As a result, there could be improvements in the change of direction ability following PT (*Sheppard & Young, 2006*; *Thomas, French & Hayes, 2009*). Moreover, PT can enhance eccentric strength in the thigh muscles, which is crucial for deceleration during impulsive movements (*Sheppard & Young, 2006*). This enhancement enables a rapid transition from eccentric to concentric muscle action in the leg extensors, thus facilitating direction changes (*Asadi et al., 2017*). Furthermore, a high

sprint speed over short distances is closely correlated with change of direction ability (*Nimphius, McGuigan & Newton, 2010*; *Pereira et al., 2018*), and PT often includes exercises or movements that involve quickly decelerating and then explosively accelerating in the opposite direction (*Ramírez-Campillo, Andrade & Izquierdo, 2013*; *Möck & Rosemann, 2023*). Overall, the findings presented in this section suggest that PT is useful for improving the change of direction ability of racket sport athletes.

## Reaction time

A reduction of milliseconds in reaction time can enable players to execute quicker changes in direction, reach certain points on the court more rapidly, and respond to faster balls during gameplay (*Salonikidis & Zafeiridis, 2008*). Our meta-analysis showed that PT had a positive effect on reaction time. However, it is difficult to compare our findings to earlier evidence because the effect of PT on reaction time has not been studied to a great extent in the literature. For ball sports (*e.g.*, racket sports), the capacity to respond to visual stimuli is linked to the organization of the motor control system, which relies on the information provided by the perceptual system (*McLeod, 1987*). Reactive actions in sports encompass a range of offensive and defensive maneuvers, involving accelerated movement, sudden stops, and rapid decelerations, all of which depend on the player's speed of response to stimuli (*Yildiz et al., 2020*; *Prabhu, Kulkarni & Palekar, 2022*). Therefore, enhanced reaction time can be an anticipated outcome of PT, as it serves as a means to enhance the sensory-motor system (*Turgut et al., 2019*). Researchers have observed a positive relationship between change of direction ability and reaction time, suggesting that players with better change of direction speed also tend to react more quickly (*Homoud, 2015*; *Moradi & Esmaeilzadeh, 2015*). Notably, PT has been shown to yield improvements in change of direction ability, as discussed elsewhere in this review (*Asadi et al., 2017*; *Falch, Guldteig Rædergård & van den Tillaar, 2019*). Additionally, some studies have assessed the impact of PT on reaction time in other non-racket sports. For example, a twelve-week ballistic-six exercise program was shown to be effective in enhancing reaction time among volleyball players (*Turgut et al., 2019*). *Chottidao et al. (2022)* discovered that eight weeks of PT improved reaction time in boxers. Therefore, PT can be considered an effective training modality for enhancing the reaction time of racket sport athletes.

## Balance and flexibility

In athletes, balance is important for preventing sports injuries and achieving successful performance in sports skills (*Hrysomallis, 2007, 2011*). However, among the studies included in our review, only one specifically investigated the balance capacity of badminton players (*Alikhani et al., 2019*). This research revealed that eight weeks of PT significantly improved the dynamic balance of badminton players. Previously, a meta-analysis conducted by *Ramachandran et al. (2021)* reported moderate effects of PT on the balance performance of healthy participants. PT offers the benefit of utilizing both the propelling (concentric) and braking (eccentric) phases of the SSC in comparison to other training approaches (*Markovic & Mikulic, 2010*). This unique aspect of PT leads to improvements in quickness, strength, and power (*Taube, Leukel & Gollhofer, 2012*).

The importance of adequate lower-extremity muscle strength and power for maintaining or restoring balance in daily activities (*e.g.*, falling) and sports-related tasks (*e.g.*, jump-landing) highlights the significance of PT as an effective means to enhance balance (*Blackburn et al., 2000*; *Muehlbauer, Gollhofer & Granacher, 2015*; *Vetrovsky et al., 2019*).

Maintaining a good level of flexibility not only aids in executing movements effectively but also helps in avoiding injuries (*Dantas et al., 2011*). However, in our review, only one study examined the flexibility of badminton players (*Albayati & Kaya, 2022*). The findings from this study demonstrated a non-significant change in the flexibility of the badminton players after twelve weeks of PT. Notably, having a sufficient range of motion in joints is highly significant in racket sports, particularly when executing specific motor gestures that require maximum movement, such as the serve in tennis (*Chang, Liu & Chang, 2016*) or the smash in badminton (*Zhang et al., 2016*). Theoretically, PT is beneficial for flexibility. This can be attributed to the activation of the SSC, which combines stretching with muscular contraction. This mechanism potentially explains the positive effects of the eccentric component (*O'Sullivan, McAuliffe & DeBurca, 2012*).

Nonetheless, there is a lack of data demonstrating the effectiveness of PT on balance and flexibility tasks in racket sports players at this time. PT was reported to have greater effects on balance and flexibility in prior studies, but these interventions were tested on handball players (*Hammami et al., 2020*), basketball players (*Arazi & Asadi, 2011*), or soccer players (*Ramírez-Campillo et al., 2015*). The demands placed on those athletes differ from those on racket sports players (*Lees, 2003*), implying that their physical capacities and regular training sessions are not comparable to those of racket sports players. Therefore, more research is needed to confirm the impact of PT on balance and flexibility in racket sports athletes.

## Limitations

This review has a few limitations that need to be highlighted. Firstly, the current study did not investigate the impact of the PT program on other physical fitness parameters, such as endurance and coordination. Future research in this cohort (*i.e.*, racket sports players) should address these aspects to acquire a deeper understanding. Moreover, due to the scarcity of research (<3) for at least one programming parameter, it was impossible to conduct additional analyses on PT frequency, length, and total sessions for all physical fitness performance measures. Secondly, while the investigations included in the analysis did not explicitly mention any adverse health events related to the PT interventions, it is uncertain if the researchers made a comprehensive effort to document all potential adverse responses. Consequently, to enhance our understanding of the safety aspects associated with this form of training, future research should describe any discomfort, injuries, or adverse events that may be linked to PT. Thirdly, the majority of participants (80%) in the included studies were male players, indicating a need for research involving female athletes on this topic. As a result, the generalizability of our findings is somewhat limited, revealing a gap in the existing literature. Despite the aforementioned limitations, our systematic review with meta-analysis offers a novel and noteworthy value to the current body of

knowledge, shedding light on the advantages of PT in enhancing various essential aspects of physical fitness in racket sports athletes.

### Practical applications

The findings of this review have applications for coaches and practitioners. Firstly, it is suggested that PT is part of the training for racket sport athletes, rather than relying solely on high-volume routine training, to improve important indices of physical fitness, specifically muscle power, muscle strength, sprint speed, change of direction ability, and reaction time. Secondly, researchers are strongly encouraged to undertake well-designed studies investigating the effects of PT in other racket sports, such as padel and squash. It is also important to explore the impact of PT specifically among female athletes. These additional studies are crucial to further validate and reinforce the conclusions drawn in this analysis. Thirdly, the utilization of PT is a cost-effective alternative compared to other training strategies, as it requires no or minimal equipment. Typically, PT involves engaging in drills that utilize the athlete's body weight as a load (*Ramirez-Campillo et al., 2020b*). This makes it a convenient and accessible training approach for athletes, allowing them to easily incorporate it into their routines.

## CONCLUSIONS

This meta-analysis reveals that PT can be implemented as an effective form of training to increase physical fitness in racket sport athletes, including muscle power, muscle strength, sprint speed, change of direction ability, and reaction time. These findings were observed among racket sport athletes involving badminton players, tennis players, and table tennis players. While it is unable to provide precise recommendations for each training parameter, the current review has presented some preliminary data on evidence-based recommendations, such as a training period of three to twelve weeks, a training frequency of one to three sessions per week, a training volume of one to three sets per exercise, and seven to thirty repetitions per set. Future studies are required to clarify the optimal doses and examine interactions among training variables to further promote the physical fitness of this specific population.

### Funding

The authors received no funding for this work.

### Competing Interests

The authors declare that they have no competing interests.

### Author Contributions

- Nuannuan Deng conceived and designed the experiments, performed the experiments, analyzed the data, prepared figures and/or tables, authored or reviewed drafts of the article, and approved the final draft.
- Kim Geok Soh conceived and designed the experiments, authored or reviewed drafts of the article, and approved the final draft.
- Borhannudin Abdullah conceived and designed the experiments, analyzed the data, authored or reviewed drafts of the article, and approved the final draft.
- Dandan Huang performed the experiments, analyzed the data, prepared figures and/or tables, authored or reviewed drafts of the article, and approved the final draft.

## Data Availability

The date used for meta-analysis are available in the Supplemental File.

## Supplemental Information

Supplemental information for this article can be found online at http://dx.doi.org/10.7717/peerj.16638#supplemental-information.

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
