# Peer review of "Effects of plyometric training on measures of physical fitness in racket sport athletes: a systematic review and meta-analysis"

_PeerJ, doi:10.7717/peerj.16638_

## Round 0.1 · original submission · Major Revisions

Please review the reviewers' comments.

Reviewer 1 ·

Basic reporting

・Clear and unambiguous, professional English used throughout
Line 67: Please switch “plyometric training” to “PT.”

Lines 81, 160, 228: Please use the term "sex" instead of "gender." "Sex" refers to a physical or biological attribute, whereas "gender" is a sociopsychological term.

Lines 81, 157, 315, etc.: The term "agility" may not be entirely appropriate. According to Young et al. (2002, PMID: 12094116; 2021, doi: 10.1186/s40798-021-00304-y), agility encompasses various factors such as perceptual and decision-making elements, change of direction speed, and more. While your definition (Lines 473-475) is accurate, the studies included in this review employed the Illinois test, t-test, and shuttle run to evaluate "agility" (but not true agility). These assessments solely involve running a predefined course and do not gauge responsiveness to external stimuli. Consequently, I recommend reviewing the usage of "agility" and "change of direction ability" in the Reaction time subsection of the Discussion and, if necessary, systematically replacing "agility" with "change of direction ability" throughout the entire manuscript.

Lines 91-93: It seems redundant in its expression. I recommend rephrasing to, for example, “a systematic review with a meta-analysis can reveal the limitations and gaps within the existing literature on PT.”

Line 139: You mentioned "lasting longer than 2 weeks," but it appears to be an error. Please correct it to "lasting shorter than 2 weeks."

Line 248: Please review the number of studies. As far as I see Table 1, nine studies obtained a score of 4 or 5.

Line 351: There are two spaces in “I2 = 0.0%.”

Lines 356-357: I recommend placing this sentence following “high levels (I2 = 0.0%–76.5%).”

Lines 403-431: I understood the mechanism behind the improvement of muscle strength through PT. However, I am curious about the reason hops and jumps led to an enhancement in grip strength (Albayati et al., 2022). While it may not be directly related to the main topic, there could be significant practical applications from an applied perspective. Thus, I recommend considering an addition that delves into the mechanism behind the improvement in grip strength.

Line 408: Please switch “effective technique” to “effective training modality (or type).”

Lines 409 and 424: I recommend eliminating muscle strength “performance.”

Line 442: Please switch “plyometrics improved the performance of sprint speed” to “PT improved the sprint speed (or time, or performance).”

Line 448: "Muscular activation capacity" doesn't seem to be a commonly used term. I suggest revising it to a different term like "muscular activity" or "muscle activation level."

Line 449: Please replace the term "muscular factors" with alternative terms such as "physiological factors." The term "muscular factors" may encompass various elements including neuromuscular, physiological, and musculotendinous factors.

Lines 452-453: I recommend removing the sentences "Therefore, ... expected," as they might be unnecessary information.

Line 454: I would like to suggest considering a paragraph break at Line 454 “Futhermore,”. This change would help delineate the different aspects you are discussing and make the content easier to follow for readers.

Line 497: Please delete PT “training.”

Lines 531-532: I recommend revising the expression “This research revealed that after eight weeks of PT, the badminton players’ dynamic balance significantly improved” to “This research revealed that eight weeks of PT significantly improved the dynamic balance for the badminton player.”

Line 534: Could you please clarify whether you are referring to "PT", "plyometric exercises" or “plyometric drill”? Please ensure that the distinction between the terms "plyometric training" (PT, chronic exercise) and "plyometric exercises or drill" (acute) is accurately maintained throughout the other sections as well.

・Literature references, sufficient field background/context provided
The reference style in PeerJ adheres to an alphabetized reference list format. Please double-check the order of your references to align with this format. Furthermore, ensure that all studies included in the present review are properly cited as references. Notably, some studies (Chou, 2022; Kaabi et al., 2022; Panda et al., 2022, etc.) have not been cited.

・Professional article structure, figures, tables. Raw data shared.
The captions of Figures 2-6: Please revise “Hedges’s g” to “Hedges’ g”

Table 2: Please revise “48 weeks.”

Table 3: You mentioned maximum isometric force (Fmax, leg), but could you specify the type of Fmax the studies are measuring (knee extension or flexion? Or hip?)? Moreover, kindly review the formatting (×, -, and spacing) of the column pertaining to the type of exercise.

Experimental design

Line 132: While you have included studies that implemented a PT intervention lasting ≥ 2 weeks, it's important to consider that a 2-week intervention (3 weeks in this review) might not provide adequate time for significant neuromuscular, physiological, and biomechanical adaptations to occur. If there are references supporting the idea that a 2-week intervention leads to these adaptations, please provide them. To my knowledge, neuromuscular adaptations are typically induced by interventions lasting ≥ 4 weeks (Maeo et al., 2018, DOI: 10.1249/MSS.0000000000001611).

Validity of the findings

None.

Additional comments

It appears that there might be an excessive focus on future research directions in the Discussion section. While focusing on the future direction is important, it could be beneficial to shift the emphasis toward highlighting the findings obtained from this study. I recommend revising the Discussion to place a stronger emphasis on the outcomes of this research.

Reviewer 2 ·

Basic reporting

Overall, it’s an interesting and well-documented study. Only have a few minor recommendations.



Abstract:

Recommend being clear what “ES” is.

“The age of athletes and training variables did not influence the effect of PT on muscle power performance.” --- Recommend presenting quantitative evidence (in addition to qualitative statement).



Main text:

Line 202: “The random-effects model was applied to account for differences across trials that may influence the PT effects” --- Why not fixed-effect analysis? Or why not presenting both (e.g., with one as the base case and the other as sensitivity analyses)?

Experimental design

No comment - fairly standard design

Validity of the findings

No comment

Additional comments

No comment

Reviewer 3 ·

Basic reporting

Grammar and anguage ccan be improved

Experimental design

Line 138: I think the authors mean acute interventions lasting less than 2 eeeks (not longer)

Line 188: Imprecision should not be based solely on the statistical heterogeneity as I2, as it can paint an exaggerated picture in many cases [1].

Lines 206: Do the effecr sizes fall within the range or are greater than the range?





1. Zeng L, Brignardello-Petersen R, Hultcrantz M, Mustafa RA, Murad MH, Iorio A, Traversy G, Akl EA, Mayer M, Schünemann HJ, Guyatt GH. GRADE Guidance 34: update on rating imprecision using a minimally contextualized approach. J Clin Epidemiol. 2022 Oct;150:216-224. doi: 10.1016/j.jclinepi.2022.07.014. Epub 2022 Aug 4. PMID: 35934265.

Validity of the findings

Line 248 to 250: There are a total of 14 studies. But, here 8 are said to be of moderate quality, and 5 of high quality

The GRADE assessment should be mentioned in the results, so that the process can be followed

There is no mention of publication bias or its regression test (or other tests), or visual inspection. This needs to be added.

I feel that the authors have understated the importance of their results too much in the conclusion. They can need not do so.

---

## Round 0.2 · Minor Revisions

Looking forward to your revised manuscript.

Reviewer 1 ·

Basic reporting

Lines 24-25: Which effect size corresponds to the younger group? And, please clarify the meaning of the term 'training variables.
Lines 51-52: I recommend eliminating “while also serving as a means of injury prevention,” as this review does not focus on or discuss injury risks.
Line 166: You used “´” for Stojanović, but his name should be Stojanović.
Lines 417-418: I think the following phrasing is preferred: "The strength improvements induced by PT.”
Lines 458-459: The article should clarify through which changes in SSC the sprint time improves.
Line 459: Please delete sprint speed “time.”
Lines 494-497 & lines 521-522, 561-562: This information should be included in the Method section, as it explains the methodological definition of the change of direction ability (lines 494-497). Alternatively, it may be better to consider removing this sentence.
Lines 500-503: This sentence would be redundant. It should be shortened.
Line 606: Please revise “be” to “is.”

Experimental design

no comment

Validity of the findings

no comment

Additional comments

The Discussion section contains substantial redundancy in the text. These sentences can be made less redundant by either deleting one or by combining them.
For example, lines 369-370 would have the same meaning as lines 377-378.
・Lines 381-382 and 382-385.
・Lines 423-425, 425-426, and 426-428, 433-435.
・Lines 459-460 and 460-461.
・Lines 509-511, 511-512, and 513-515.
I believe this comment may apply to other sections as well (lines 43-49), so please make revisions as necessary.

Reviewer 2 ·

Basic reporting

Just one more comment about the response to one of my initial comments “Line 202: “The random-effects model was applied to account for differences across trials that may influence the PT effects” --- Why not fixed-effect analysis? Or why not presenting both (e.g., with one as the base case and the other as sensitivity analyses)?”



The authors didn’t reply when fixed-effect analyses were not conducted. One the of main reasons I asked was because the reported I2 estimates consistently show a low level of heterogeneity, which appeared to support the use of fixed-effect analyses over random-effect analyses.



If fixed-effect analyses were not considered, then at least, the manuscript could have been a bit clearer as to (1) the rationale of not considering fixed-effect analyses and (2) interpretation of the estimated I2.

Experimental design

No further comment

Validity of the findings

No further comment

Additional comments

No further comment

---

## Round 0.3 · accepted · Accept

Thank you for responding to all reviewer comments.

Reviewer 1 ·

Basic reporting

No comment.

Experimental design

No comment.

Validity of the findings

No comment.

Additional comments

No comment.

Reviewer 2 ·

Basic reporting

The prior comments have been appropriately addressed. I have no further comment.

Experimental design

The prior comments have been appropriately addressed. I have no further comment.

Validity of the findings

The prior comments have been appropriately addressed. I have no further comment.